# SAM-Competitive EZH2-Inhibitors Induce Platinum Resistance by EZH2-Independent Induction of ABC-Transporters

**DOI:** 10.3390/cancers15113043

**Published:** 2023-06-03

**Authors:** Elisabeth Groß, Ralf-Axel Hilger, Franziska Lea Schümann, Marcus Bauer, Alyssa Bouska, Christian Rohde, Edith Willscher, Jana Lützkendorf, Lutz Peter Müller, Bayram Edemir, Thomas Mueller, Marco Herling, Mascha Binder, Claudia Wickenhauser, Javeed Iqbal, Guido Posern, Thomas Weber

**Affiliations:** 1Department of Hematology and Oncology, Martin-Luther-University Halle-Wittenberg, 06120 Halle (Saale), Germany; 2West German Cancer Center, University Hospital Essen, 45147 Essen, Germany; 3Institute of Pathology, Martin-Luther-University Halle-Wittenberg, 06112 Halle (Saale), Germany; 4Department of Pathology and Microbiology, University of Nebraska Medical Center, Omaha, NE 68198, USA; 5Department of Hematology, Oncology and Rheumatology, Heidelberg University Hospital, 69120 Heidelberg, Germany; 6Department of Hematology, Cell Therapy, Hemostaseology, University of Leipzig, 04103 Leipzig, Germany; 7Institute for Physiological Chemistry, Martin-Luther-University Halle-Wittenberg, 06114 Halle (Saale), Germany

**Keywords:** chemotherapy, drug resistance, oxaliplatin, EZH2, EZH2 inhibitors, combination therapy, mature T-cell lymphoma, T-cell leukemia

## Abstract

**Simple Summary:**

The histone lysine methyltransferase EZH2 is frequently altered in lymphoid tumors. Its overexpression or mutation is associated with tumor progression and resistance to chemotherapy. This makes it an attractive target for inhibition especially in combination treatments with established chemotherapeutics with the goal of overcoming chemotherapy resistance. However, the impact of antagonistic effects in rationally designed drug combinations remains poorly understood necessitating thorough investigation. In the current study, we show that the combinational treatment with SAM-competitive EZH2 inhibitors leads to platinum resistance due to increased platinum efflux. On a molecular level, we have discovered off-target effects leading to the upregulation of proteins that are associated with chemotherapy resistance. Our findings underline the need for detailed studies of combination therapies in order to rule out adverse effects of rational therapeutic approaches.

**Abstract:**

T-cell lymphomas are heterogeneous and rare lymphatic malignancies with unfavorable prognosis. Consequently, new therapeutic strategies are needed. The enhancer of zeste homologue 2 (EZH2) is the catalytic subunit of the polycomb repressive complex 2 and responsible for lysine 27 trimethylation of histone 3. EZH2 is overexpressed in several tumor entities including T-cell neoplasms leading to epigenetic and consecutive oncogenic dysregulation. Thus, pharmacological EZH2 inhibition is a promising target and its clinical evaluation in T-cell lymphomas shows favorable results. We have investigated EZH2 expression in two cohorts of T-cell lymphomas by mRNA-profiling and immunohistochemistry, both revealing overexpression to have a negative impact on patients’ prognosis. Furthermore, we have evaluated EZH2 inhibition in a panel of leukemia and lymphoma cell lines with a focus on T-cell lymphomas characterized for canonical EZH2 signaling components. The cell lines were treated with the inhibitors GSK126 or EPZ6438 that inhibit EZH2 specifically by competitive binding at the *S*-adenosylmethionine (SAM) binding site in combination with the common second-line chemotherapeutic oxaliplatin. The change in cytotoxic effects under pharmacological EZH2 inhibition was evaluated revealing a drastic increase in oxaliplatin resistance after 72 h and longer periods of combinational incubation. This outcome was independent of cell type but associated to reduced intracellular platinum. Pharmacological EZH2 inhibition revealed increased expression in SRE binding proteins, SREBP1/2 and ATP binding cassette subfamily G transporters ABCG1/2. The latter are associated with chemotherapy resistance due to increased platinum efflux. Knockdown experiments revealed that this was independent of the EZH2 functional state. The EZH2 inhibition effect on oxaliplatin resistance and efflux was reduced by additional inhibition of the regulated target proteins. In conclusion, pharmacological EZH2 inhibition is not suitable in combination with the common chemotherapeutic oxaliplatin in T-cell lymphomas revealing an EZH2-independent off-target effect.

## 1. Introduction

Enhancer of zeste homologue 2 (EZH2) is part of the polycomb repressive complex 2 (PRC2) and canonically responsible for the inactivation of gene transcription by trimethylation of lysine 27 of histone 3 (H3K27me3) [1]. Additionally, PRC2-independent (non-canonical) signaling by EZH2 is known and canonical pathways can partially be undertaken by the related protein enhancer of zeste homologue 1 (EZH1) [2,3]. EZH2 was found to be overexpressed in a variety of tumor entities and is thought to be involved in cancer progression of T-cell leukemia [4]. Consequently, a variety of EZH2 inhibitors have been developed that have already entered clinical trials [5].

Peripheral mature T-cell Non-Hodgkin’s Lymphomas (MTCL) are a rare group of tumor entities with an unfavorable prognosis [6]. Due to limited therapeutic approaches, new ways of treatment are always in desperate need [7]. First line treatment often involves cyclophosphamide, doxorubicin, vincristine, and prednisolone with optional etoposide while second-line therapy is mainly based on platinum analogs, commonly the third generation platinum derivate, oxaliplatin [8,9,10].

As we and others have shown before, T-cell lymphomas overexpress EZH2, which is correlated to worse survival rates [11,12]. This gives a rationale to evaluate EZH2 inhibition as a potential new therapy of T-cell lymphomas, as a single agent but also in combination with cytotoxic agents, which is currently discussed [13].

Coherently, different types of EZH2 inhibitors were developed. Roughly, there are three main types of EZH2 inhibitors with the first-in-line being the S-adenosylhomocysteine (SAH) hydrolase inhibitor 3-Deazaneplanocin A (DZNep) that inhibits the activity of methyl transferases globally [14,15]. Secondly, there are EZH2-EED-disruptors that bind competitively at the H3K27 binding site thereby destroying the PRC2 complex [16]. Last, there are competitive *S*-adenosylmethionine (SAM) inhibitors that can be subdivided into EZH2-specific and EZH1/2 dual inhibitors [17,18]. It was already proven for different cell backgrounds, including mature T-cell lymphomas, that these types of inhibitors have promising however limited results on cell viability as single agents [19].

Efficacy of EZH2 inhibitor monotherapy in relapsed and refractory MTCL is moderate with overall response rates of the EZH1/2 inhibitor DS-3201 of 55.6% and complete response rates of 24.4% in a first-in-human study [20]. However, enhancing sensitivity to cytotoxic chemotherapy by epigenetic targeting therapies makes it an attractive combination partner. On the one hand, EZH2 inhibition was able to sensitize cells to DNA damage by doxorubicin or ionizing radiation in a variety of renewable tissues [21]. On the other hand, it was shown in acute myeloid leukemia that EZH2 inhibition leads to chemotherapy resistance against cytarabine [22]. Moreover, loss of PRC2 components is associated with increased resistance in T-cell acute lymphatic leukemia [23]. Consequently, EZH2 inhibition in T-cell lymphomas needs to be investigated thoroughly before being applied in clinical trials to rule out negative consequences.

In the current study, we have assessed EZH2 expression in MTCL and evaluated EZH2 inhibition as therapy in T-cell neoplasm cell lines in combination with established chemotherapeutics. Furthermore, we have investigated the effects on a random basis in a broad panel of cell lines from different backgrounds to elucidate possible generalized sensitizations to commonly used chemotherapeutics. Unexpectedly, we have found a drastic increase in oxaliplatin resistance through pharmacological EZH2 inhibition. We have revealed that SAM-competitive EZH2-inhibitors led to an increase in SRE binding factor (SREBF) expression in an EZH2-independent manner. We have shown that an increased ATP binding cassette subfamily G (ABCG) transporter expression is responsible for an enhanced oxaliplatin efflux.

In conclusion, while EZH2 inhibitors show promising results as well as synergistic effects with other chemotherapeutics, our data do not support the combination with platinum derivates. We have shown that pharmacological EZH2 inhibition can induce an increase in chemotherapy resistance, most prominent in oxaliplatin, which is commonly used as the second-line treatment of T-cell lymphomas. Our study underlines the importance of thorough investigations of rational drug combinations.

## 2. Materials and Methods

### 2.1. Patients and Patient Samples

The study population of the gene expression profiling cohort was described before [24]. The study included 120 cases of peripheral T-cell lymphomas, not otherwise specified (PTCL, NOS), 117 cases of angioimmunoblastic T-cell lymphomas (AITL), 14 cases of adult T-cell leukemia/lymphoma (ATLL), 48 cases of anaplastic large T-cell lymphomas (ALCL), ALK−, 31 cases of ALCL, ALK+, 23 cases of extranodal natural killer T-cell lymphomas, nasal type (NKTL), 21 cases of NKTL γδ, and 7 cases of T-cell prolymphocytic leukemia (T-PLL).

In a second clinically annotated cohort, tissue from 50 patients with MTCL was used to create tissue microarrays (TMA), as reported by Schümann et al. [12]. The study included 20 cases of peripheral T-cell lymphomas with T-helper phenotype (17 cases of AITL and 3 cases of nodal peripheral T-cell lymphomas with T-follicular helper phenotype), 8 cases of PTCL, NOS, 10 cases of ALCL, 3 cases of intestinal T-cell lymphomas, 2 cases of NKTL, and 7 cases of other subtypes (one case of mycosis fungoides, one case of Sézary syndrome, one case of subcutaneous panniculitis-like T-cell lymphoma, one case of cerebral T-cell lymphoma, one case of T-PLL and two cases of large granular lymphocytic T-cell leukemia). The performed analyses were in accordance with the guidelines of the Declaration of Helsinki and approved by the Ethics Committee of the Martin-Luther-University of Halle-Wittenberg (#2020–033, 10 May 2020).

### 2.2. EZH2 Expression Analysis

The gene expression profiling was performed as described before using HG-U133-plus2.0 arrays [24]. Detailed description of immunohistochemistry analysis is given by Schümann et al. [12]. Shortly, samples were stained for EZH2 using Bond Polymer Refine Detection Kit (DS9800-CN) and a Bond III automated immunostainer (Leica Biosystems Nussloch GmbH, Wetzlar, Germany) according to the manufacturer’s instructions. The anti-EZH2 mouse antibody (1:100, Cell Signaling, Danvers, MA, USA, Cat# 3147; RRID:AB_10694383) was used and EZH2 expression was analyzed by using the H-score [25].

### 2.3. Cell Lines

The cell lines Jurkat (Cat# ACC 282; RRID:CVCL_0065), SUPT11 (Cat# ACC 605; RRID:CVCL_2210), HH (Cat# ACC 707; RRID:CVCL_1280), and Daudi (Cat# ACC 78; RRID:CVCL_0008) were obtained from the German Collection of Microorganisms and Cell Cultures (DSMZ, Braunschweig, Germany) (13 June 2019, 2018, 2020, and 9 September 2020, respectively) and cultivated in RPMI-1640 Medium (R8758; Sigma-Aldrich, St. Louis, MO, USA) supplemented with 10% fetal bovine serum (S181B-500; Biowest, Nuaillé, France) and 1% penicillin/streptomycin (P0781; Sigma-Aldrich, St. Louis, MO, USA). Loucy (Cat# CRL-2629; RRID:CVCL_1380) was obtained from American Type Culture Collection (ATCC, Manassas, VA, USA, 2019) and cultivated in RPMI-1640 Medium supplemented with 10% fetal bovine serum and 1% penicillin/streptomycin. Oci-Ly1 (Cat# ACC 722; RRID:CVCL_1879) was obtained from DSMZ (Braunschweig, Germany) (9 September 2020) and cultivated in RPMI-1640 Medium with 20% fetal bovine serum and 1% penicillin/streptomycin. DERL-2 (Cat# ACC 531; RRID:CVCL_2016) was obtained from DSMZ (Braunschweig, Germany) (18 January 2017) and cultivated in RPMI-1640 Medium with 20% fetal bovine serum, 1% penicillin/streptomycin, and 0.02 ng/µL recombinant IL-2 (200-02, Peprotech, Hamburg, Germany). The cell line SR786 (RRID:CVCL_1711, received 30 January 2019) was cultivated in RPMI-1640 Medium with 10% fetal bovine serum and 1% penicillin/streptomycin. All cells were cultured in a humidified atmosphere at 37 °C and 5% CO_2_ and passaging did not exceed 10 times after thawing. Mycoplasma negativity was confirmed repeatedly and after the respective last experiment using MycoSPY^®^ Master Mix (M020, Biontex, Munich, Germany).

### 2.4. Cell Viability Assay

Cells were seeded at appropriate densities/100 µL/well in 96-well plates in 8 replicates. Detailed densities are listed in Appendix A. Cells were treated with inhibitors, chemotherapeutics, or in combinational approaches simultaneously at the indicated concentrations for 72 h if not stated otherwise. Cell viability was measured indirectly by fluorescence detection with a Spark^®^ multimode microplate reader (Tecan, Männedorf, Switzerland) after 2 h of Resazurin treatment (30 µg/mL, R12204; Thermo Fisher Scientific, Waltham, MA, USA). Experiments were performed in triplicates. Graphpad Prism software v8.4.3 (RRID:SCR_002798) was used for the calculation of IC_50_ values and SynergyFinder (RRID:SCR_019318) was used for the modulation and calculation of synergy scores [26]. In combinational analyses with inhibitors, stable and non-lethal concentrations (approximately 10% reduction in cell viability or published concentrations) were used: 4 µM GSK126 (HY-13470; MedChemExpress, Sollentuna, Sweden), 4 µM EPZ6438 (HY-13803; MedChemExpress, Sollentuna, Sweden), 125 nM DZNep (Cay-13828; Biomol, Hamburg, Germany), 1 µM Fatostatin (HY-14452; MedChemExpress, Sollentuna, Sweden), or 30 µM Glibenclamide (HY-15206; MedChemExpress, Sollentuna, Sweden) [27].

### 2.5. Apoptosis Assay

Cells were seeded at 500 k cells/mL and incubated with 4 µM GSK126, oxaliplatin at half maximal reduction in viability or both for 24, 48, or 72 h before being collected by centrifugation and cleared from supernatant. The pellets were washed with PBS and stained using FITC Annexin V Apoptosis Detection Kit with 7-AAD (640922; Biolegend, San Diego, CA, USA) according to the manufacturer’s instructions. Cells were analyzed by flow cytometry using BD LSR Fortessa™ Cell Analyzer (BD Biosciences, Vaud, Switzerland) and evaluated by BD Life Sciences FlowJo™ v10.6.1. Experiments were performed thrice and the exemplary gating strategy is shown in Appendix A. Annexin V-positive, 7-AAD-negative cells were considered apoptotic and Annexin V-positive, 7-AAD-positive cells were considered late necrotic.

### 2.6. Cell Cycle Assay

Cells were seeded at 500 k cells/mL and incubated with 4 µM GSK126, oxaliplatin at half maximal reduction in viability or both for 24, 48, or 72 h before being collected by centrifugation and cleared from supernatant. The pellets were washed with PBS and fixed with 70% ice-cold ethanol for at least 24 h at 4 °C. The ethanol was removed and cells were washed with PBS. The cells were incubated in 1 mg/mL RNase (EN0531; Thermo Fisher Scientific, Waltham, MA, USA) at 37 °C for 30 min. The pellets were stained with DAPI (268298; Sigma-Aldrich, St. Louis, MO, USA), directly analyzed by flow cytometry using BD LSR Fortessa™ Cell Analyzer (BD Biosciences, Vaud, Switzerland), and evaluated by BD Life Sciences FlowJo™ v10.6.1. Experiments were performed in triplicates and the exemplary gating strategy is shown in Appendix A.

### 2.7. Mass Spectrometry with Inductive Coupled Plasma (ICP-MS)

ICP-MS is based on the vaporization, atomization, and ionization of a sample in plasma for subsequent mass spectrometric analysis and was applied as described before [28,29]. The plasma is a very efficient ion source and provides the technique with a high sensitivity allowing the determination of most elements at trace concentrations and has a sensitivity of at least 5 ppt for platinum. Cells were incubated with oxaliplatin at half maximal reduction in viability as single agent or in combination with 4 µM GSK126, 1 µM Fatostatin, 30 µM Glibenclamide, or in combination for 24, 48, or 72 h before being collected by centrifugation and cleared from supernatant. Pellets were incubated in 1% HNO_3_ at 70 °C over night. Platinum content was measured by inductively coupled plasma mass spectrometry using Bruker 820-MS ICP Mass Spectrometer (Bruker Daltonik, Billerica, MA, USA) with Agilent ICP-MS Expert software v2.1 and expressed as ng of platinum per 10^6^ cells. A single ICP-MS measurement represents the average of 20 scans per replicate within 5 replicates from the same liquid sample, with an error <5%. The platinum concentrations that we report at each time point were averaged across four series of cultures, ensuring that the values are correctly scaled to account for cell population differences and dilutions. Standard curves were generated by using aqueous serial dilutions of stock solutions traceable back to standard reference material from the National Institute of Standards and Technology. For comparison, the values were normed to the respective oxaliplatin monotherapy control at each given time point. Experiments were performed thrice.

### 2.8. Transduction

Cells were seeded at 200 k cells/500 µL/well in 24-well plates in RPMI-1640 Medium supplemented with 10% fetal bovine serum, 1% penicillin/streptomycin, and polybrene (10 µg/mL, sc-134220; Santa Cruz Biotechnology, Dallas, TX, USA). EZH2 knockdown virus particles were obtained from Sigma-Aldrich (St. Louis, MO, USA) (sequences listed in Appendix A) and 3 µL of virus was added per well. Cells were centrifuged at 600× *g* for 45 min. After 24 h, media were changed to RPMI-1640 Medium supplemented with 10% fetal bovine serum and 1% penicillin/streptomycin and after 48 h selection by puromycin (1 µg/mL, P9620-10ML; Sigma-Aldrich, St. Louis, MO, USA) was applied.

### 2.9. Cell Treatment for Expression Analysis

Cells were seeded at 500 k cells/mL and incubated with inhibitors, oxaliplatin or in combination for 72 h before being collected by centrifugation and cleared from supernatant. The pellets were washed with PBS and stored at −80 °C. The following non-lethal stable concentrations were used: 4 µM GSK126, 4 µM EPZ6438, 125 nM DZNep, or oxaliplatin at 10% reduction in viability.

### 2.10. Western Blot

Protein isolation and Western blot analysis was carried out as described before with some modifications [30]. Briefly, cells were collected by centrifugation and cleared from supernatant. Total protein lysate were isolated using RIPA buffer (R0278; Sigma-Aldrich, St. Louis, MO, USA) with protease inhibitor mix (143 μL/mL, 11836170001; Roche, Basel, Switzerland) and Benzonase (250 Units/mL, E1014-24KU; Sigma-Aldrich, St. Louis, MO, USA). Histone isolation was carried out using the Abcam (Cambridge, UK) histone extraction kit (ab113476) according to the manufacturer’s instructions. Lysates were separated using SDS-PAGE with 4–12% Novex™ Bis-Tris gradient gel (NP0322BOX; Thermo Fisher Scientific, Waltham, MA, USA) with Novex™ NuPAGE™ MES SDS running buffer (NP0002; Thermo Fisher Scientific, Waltham, MA, USA) before being blotted onto 0.2 μm nitrocellulose membrane (10600001; GE Healthcare, Chicago, IL, USA). Unspecific binding sites were blocked with 2% bovine serum albumin or 5% skim milk solution. The following antibodies were used: Anti-EZH2 mouse (1:1000; Cell Signaling, Danvers, MA, USA; Cat# 3147; RRID:AB_10694383), anti-GAPDH rabbit (1:10000; Cell Signaling, Danvers, MA, USA; Cat# 14C10; RRID:AB_561053), anti-ABCG1 rabbit (1:500; Novus Biologicals, Centennial, CO, USA; Cat# NB400-132; RRID:AB_10125717), anti-ABCG2 [EPR20080] rabbit (1:1000; Abcam, Cambridge, UK; Cat# ab207732), anti-SREBP1 mouse (1:1000; BD Biosciences, Vaud, Switzerland; Cat# 557036; RRID:AB_396559), anti-SREBP2 mouse (1:1000; BD Biosciences, Vaud, Switzerland; Cat# 557037; RRID:AB_396560), anti-H3K27me3 rabbit (1:2000; Cell Signaling, Danvers, MA, USA; Cat# 9733; RRID:AB_2616029) and anti-H3 rabbit (1:10,000; Abcam, Cambridge, UK; Cat# ab1791; RRID:AB_302613). As secondary antibody, Peroxidase-conjugated AffiniPure F(ab’)_2_ Fragment goat-anti-rabbit IgG (1:5000; Jackson Immunoresearch, Ely, UK; Cat# 111–036-047; RRID:AB_2337945) and Peroxidase-conjugated AffiniPure F(ab’)_2_ Fragment goat-anti-mouse IgG (1:5000; Jackson Immunoresearch, Ely, UK; Cat# 115-036-062; RRID:AB_2307346) were used. Primary antibodies were diluted in skim milk or bovine serum albumin solution according to the manufacturer’s instruction, incubated over night at 4 °C, and afterwards washed in PBST. Horse radish peroxidase-coupled secondary antibodies were diluted in skim milk and incubated for 1 h at room temperature. The membrane was washed in PBST before being incubated with ECL™ Prime Western Blotting Detection Reagent (GERPN2236; GE Healthcare, Chicago, IL, USA) and signals were detected on the ChemiDoc Imager detecting system (Bio-Rad, Hercules, CA, USA). Western blots were performed from at least 3 individual cell lysates per sample and a representative blot is shown. Quantitative analyses were performed using National Institutes of Health ImageJ v1.51 (RRID:SCR_003070) for band intensity calculation. Values were normed to the respective housekeeping control before being normed to the control cell line. Whole uncropped Western blots can be found in Appendix A.

### 2.11. cDNA Synthesis and RT-qPCR

RNA was isolated using peqGOLD TriFast™ (30-2010, VWR, Radnor, PA, USA) according to the manufacturer’s instructions. cDNA was synthesized from 2 µg RNA using MMLVRT protocol (28025-013; Thermo Fisher Scientific, Waltham, MA, USA). Target gene expression was analyzed by RT-qPCR using SYBR™ Select Master Mix for CFX (4472952; Applied Biosystems, Waltham, MA, USA) according to the manufacturer’s instructions using 60 ng cDNA and a primer concentration of 250 nM. Data were acquired using CFX384 Touch Real-Time PCR detection system (Bio-Rad, Hercules, CA, USA) with Bio-Rad CFX Manager Software v3.1 (RRID:SCR_017251) and quantified using the comparative 2^−ΔΔCT^ method as described [31]. The housekeeping gene TBP was used for normalization. Experiments were performed from 3 individual cell lysates. Primer sequences are listed in Appendix A.

### 2.12. RNA-Sequencing of Treated Cell Culture Samples

RNA was isolated using peqGOLD TriFast™ according to the manufacturer’s instructions and analyzed by Genewiz (Leipzig, Germany) from two independent experiments of the cell lines HH, DERL2, Jurkat, and Loucy. Quality control, sequencing, and basic bioinformatics were executed by Genewiz as service. RNA samples were quantified using Qubit 4.0 Fluorometer (Life Technologies, Carlsbad, CA, USA) and RNA integrity was checked with RNA Kit (DNF-471; Agilent Technologies, Santa Clara, CA, USA) on Agilent 5300 Fragment Analyzer (Agilent Technologies, Santa Clara, CA, USA) with a mean quality score of 35.95. RNA-sequencing library preparation was prepared using NEBNext Ultra II Directional RNA Library Prep Kit for Illumina following the manufacturer’s instructions (E7760; NEB, Ipswich, MA, USA). Sequencing libraries were validated using the NGS Kit on the Agilent 5300 Fragment Analyzer and quantified by Qubit 4.0 Fluorometer. The sequencing libraries were multiplexed and loaded on the flowcell on the Illumina (San Diego, CA, USA) NovaSeq 6000 instrument according to the manufacturer’s instructions. The samples were sequenced using a 2 × 150 Pair-End (PE) configuration v1.5 with 93.95% of bases having a read depth >30. Image analysis and base calling were conducted by the NovaSeq Control Software v1.7 on the NovaSeq instrument. Raw sequence data were converted into fastq files and de-multiplexed using Illumina bcl2fastq v2.20 (RRID:SCR_015058). One mismatch was allowed for index sequence identification. Sequence reads were trimmed to remove possible adapter sequences and nucleotides with poor quality using Trimmomatic v0.36 (RRID:SCR_011848). The trimmed reads were mapped to the *Homo sapiens* reference genome available on ENSEMBL using the STAR aligner v2.5.2b (RRID:SCR_004463). Unique gene hit counts were calculated using the feature Counts from the Subread package v1.5.2 (RRID:SCR_009803). Only unique reads that fall within exon regions were counted. Raw counts of all samples were combined to a matrix, gene names were annotated using biomaRt (RRID:SCR_019214) [32]. We omitted genes whose sum of counts over all samples was 0. Normalization, size correction, and transformation to log2 was performed with package deseq2 (RRID:SCR_015687) [33]. Differentially expressed genes were calculated with lfcshrink from deseq2 on log2 transformed data. Volcano Plot was visualized with package EnhancedVolcano. Gene Set Enrichment Analysis on GeneSets was performed with enrichR (RRID:SCR_001575) [34]. All analysis steps were performed in RStudio with R v4.1.2 (RRID:SCR_000432).

### 2.13. Statistics

Basic statistics were performed using Graphpad Prism software v8.4.3 applying two-sided, unpaired Student’s *t*-test if not stated otherwise. For normed analyses, Welch’s correction was applied. For the comparisons of expression data between samples, two-sided, unpaired Student’s *t*-test (2 groups) or one-way ANOVA test (>2 groups) were applied. Survival data were illustrated by Kaplan–Meier-plots and group comparisons were performed by log-rank statistics.

## 3. Results

### 3.1. MTCL Overexpress EZH2, but Selective EZH2 Inhibitors Show No Reduction of Cell Viability

The EZH2 expression was analyzed in two cohorts of MTCL samples on gene expression or protein level and differences to healthy lymphoid samples were assessed. In a cohort of MTCL with different cellular backgrounds (*n* = 381), EZH2 gene expression profiling was analyzed and compared to normal lymphoid control tissue (*n* = 14). Tumor tissues were found to have significantly higher EZH2 expression than healthy control (tumor vs. control: median EZH2 expression 11.07 (IQR = 1.02) vs. 10.24 (IQR = 1.13); *p* < 0.0001) (Figure 1A). In the same cohort, univariate analysis revealed high EZH2 expression to be associated with poorer overall survival (OS) rates including all tumor entities (EZH2^low^ vs. EZH2^high^: median OS 2.45 vs. 1.42 years; *p* = 0.0096) (Figure 1B).

Apart from gene expression, EZH2 protein expression was assessed. In a cohort of MTCL (*n* = 50), EZH2 protein expression was compared to healthy lymph node samples (*n* = 10) and found to be significantly elevated in tumor tissue (tumor vs. control: median EZH2 H-score 85.0 (IQR = 115.0) vs. 30.0 (IQR = 28.0); *p* = 0.0063) (Figure 1C). Furthermore, high EZH2 expression was associated with poorer OS rates in nodal MTCL (EZH2^low^ vs. EZH2^high^: median OS 113 vs. 35 months; *p* = 0.0456) in univariate analysis (Figure 1D).

The validated overexpression of EZH2 in MTCL provides a rationale for EZH2 inhibition as a possible targeted therapy in these entities. While others could show first promising results by EZH2 inhibition, we could not validate a significant reduction in cell viability by GSK126 and EPZ6438 at clinically relevant concentrations in a well-characterized panel of T-cell neoplasm cell lines, each with known PRC2 mutation status and H3K27 trimethylation level (Figure 1E; Appendix A). The panel included one cell line with cutaneous T-cell lymphoma (CTCL), one cell line with hepatosplenic T-cell lymphoma (HSTCL) and two cell lines with a T-cell acute lymphoblastic leukemia (T-ALL) background. Furthermore, two cell lines from the B-cell lymphoma background with high interest in EZH2 inhibition were used for comparison as they either carry an activating EZH2 mutation or show high EZH2 expression, respectively [35,36]. The B-cell lymphoma cell lines had either diffuse large B-cell lymphoma (DLBCL) or Burkitt lymphoma (BL) origin. The data are shown exemplarily for the CTCL cell line HH and the T-ALL cell line Jurkat and can be found entirely in Appendix A. While these results prove single agent therapy not to be sufficient in EZH2 overexpressing cell lines, sensitivity modulation of established chemotherapeutics by EZH2 inhibition might still lead to advantageous combination therapies [37,38].

Therefore, we combined the commonly used chemotherapeutics doxorubicin, oxaliplatin, and cytarabin, as well as the cytotoxic and hypomethylating agent 5-azacytidin with pharmacological EZH2 inhibition by GSK126 in the same panel as before in a screening approach. The comprised maximal-found sensitivity modulation of each cell line and chemotherapeutic in combination with GSK126 is shown in Figure 1F and the detailed screening results can be found in Appendix A. The sensitivity of doxorubicin, cytarabin, and 5-azacytidin was only moderately influenced. Furthermore, a dependence of the cellular background on the type and intensity of the response was found. In contrast, oxaliplatin resistance was significantly increased independently from cell type by pharmacological EZH2 inhibition bespeaking a universal resistance mechanism of the different cell lines. As this might indicate significant side effects of EZH2 inhibitors in combination with established chemotherapy, we aimed to elucidate the induction of oxaliplatin resistance in detail.

### 3.2. SAM-Competitive EZH2 Inhibitors Increase Oxaliplatin Resistance

As the sensitivity modulation with the highest shift and the only modulation with independence from cellular background was found for oxaliplatin in the screening approach, we aimed to investigate the mechanistic changes leading to the increase in oxaliplatin resistance. Therefore, we broadened the cell line panel with one additional T-ALL cell line and one cell line with ALCL, ALK+ background. The additionally used cell lines were also well-characterized with known PRC2 mutation status and H3K27 trimethylation level (Appendix A). Furthermore, we broadened the used oxaliplatin concentration range due to cell type-dependent basal sensitivity differences. In all tested cell lines, the incubation with SAM-competitive EZH2 inhibitors but not with the SAH-hydrolase inhibitor DZNep led to a significant increase in oxaliplatin resistance which is shown representatively for the CTCL cell line HH and the T-ALL cell line Jurkat (Figure 2A) and can be found for all cell lines in Appendix A. The induction of resistance was not related to EZH2 status, H3K27 trimethylation, or cellular background indicating a universal and EZH2-independent mechanism. We validated that the reduction in oxaliplatin cytotoxicity by GSK126 treatment persisted even after 96 h (Figure 2B, Appendix A) or 7 d (Figure 2C, Appendix A) of combinational incubation.

For further studies, we focused on the cell line HH. In addition, we analyzed Jurkat in order to validate effects that might be generalized independently from the cellular background. Concentration-dependent synergy matrices of either GSK126 or EPZ6438 (Figure 2D) were composed using the internet platform Synergy Finder that showed an antagonistic effect [26]. Synergy scores were calculated according to the Highest Single Agent (HSA) model. Synergy scores lower than -10 were considered antagonistic effects underlining the reduction in oxaliplatin toxicity by pharmacological EZH2 inhibition.

In order to clearly evaluate EZH2 dependence on the induction of oxaliplatin resistance in mature T-cell lymphomas, we established an EZH2 knockdown in HH. Furthermore, the EZH2 knockdown was established in Jurkat to rule out cell line-specific effects. On the protein level, EZH2 and H3K27 trimethylation downregulation was proven (Figure 2E) and additionally, EZH2 mRNA downregulation was validated by RT-qPCR (Figure 2F). Whole uncropped Western blots from all replicates can be found in Appendix A. None of the tested shRNAs was able to induce oxaliplatin resistance. Additional GSK126 treatment, however, was able to induce oxaliplatin resistance in both mock shRNA-transduced and EZH2 knockdown cells (Figure 2G–J).

In order to elucidate the mechanism of resistance, different pathways that might be involved including induction of apoptosis, cell cycle deregulation, and drug transporter activity were analyzed. Data are shown in the exemplary cell lines HH and Jurkat (Figure 3) and all results can be found in Appendix A. As a non-lethal concentration of GSK126 was used, effects on apoptosis and necrosis or cell cycle progression by single agent incubation were neither expected nor observed (Figure 3A,B).

In all tested cell lines, oxaliplatin led to a progressive reduction in viable cells consistent with an induction of apoptotic and necrotic cells (Figure 3A, Appendix A). The effects were more prominent and earlier in T-cell lymphoma cell lines than in T-cell leukemia cell lines. Consistent with the induction of resistance, the oxaliplatin effects were lowered by additional pharmacological EZH2 inhibition by GSK126. Differences could first be observed after 48 h of incubation.

Oxaliplatin was able to induce cell cycle deregulation in all tested cell lines (Figure 3B, Appendix A). The pattern, however, differed between the cell lines. In HH, Sub-G_0_G_1_-phase was induced by oxaliplatin without any arrest in other phases. In Jurkat, however, G_2_M-arrest was induced which led to a shift to Sub-G_0_G_1_-phase. Again, the oxaliplatin effects were lowered by additional GSK126 mirroring the resistance. Once more, deregulation was first observed after 48 h of incubation and more prominent in T-cell lymphoma cell lines.

The measurement of the intracellular platinum content in the exemplary cell lines HH and Jurkat using either oxaliplatin as single agent or in combination with pharmacological EZH2 inhibition revealed lowered intracellular platinum content after 48 h of incubation (Figure 3C). The intracellular platinum content was decreased by 22.9% in HH and 18% in Jurkat (*p* < 0.006 and *p* < 0.004, respectively). The time point was consistent with the arising of differences between oxaliplatin as single agent or in combination with GSK126 from apoptosis and cell cycle progression analysis. Intriguingly, the intracellular platinum content was not reduced by EZH2 knockdown alone (Figure 3D) but could be again induced by additional pharmacological EZH2 inhibition under EZH2 knockdown (Appendix A).

In order to elucidate the involved mechanisms, we analyzed transcriptional changes under pharmacological EZH2 inhibition using mRNA-sequencing by comparing GSK126 vs. oxaliplatin regulated genes (Figure 4A). Differential gene expression and pathway regulation by pharmacological EZH2 inhibition were calculated including the four cell lines HH, DERL2, Jurkat, and Loucy as we aimed to uncover universally regulated mechanisms. Strikingly, among the significantly upregulated pathways under pharmacological EZH2 inhibition, *SREBF1* was found in each pathway (Figure 4C). The factor’s gene expression was significantly upregulated itself (Figure 4C). Consistently, the related protein Sterol Regulatory Element Binding Protein 2 (SREBP2) was reported to be regulated by SAM-competitive EZH2 inhibitors independently from EZH2 [39]. Furthermore, the transporter gene *ABCG2* was upregulated that is also known as the Breast Cancer Resistance (BCR) protein due to the induction of multi drug resistance (Figure 4B). The findings of the mRNA-sequencing were validated for the upregulated proteins as well as the closely related genes *SREBF2* and *ABCG1* using GSK126, oxaliplatin, or both simultaneously (Figure 4D). Indeed, the tested ABCG transporters as well as both SRE binding factors were upregulated by pharmacological EZH2 inhibition. Validation on protein level can be found in Appendix A and whole uncropped Western blots from all replicates can be found in Appendix A. Furthermore, expression analysis was expanded to the second SAM-competitive EZH2 inhibitor EPZ6438 and to the non-SAM-competitive EZH2 inhibitor DZNep as single agent or in combination with oxaliplatin using the same approach as before. EPZ6438 was able to induce both *ABCG1* and *ABCG2* as well as *SREBF1* and *SREBF2* expression thereby confirming a selective mechanism of SAM-competitive EZH2 inhibitors. The changes in gene expression were lower as compared to GSK126 treatment. Intriguingly, these results were not accomplished by DZNep incubation. Furthermore, subsequent oxaliplatin cell viability assay after 72 h of pharmacological EZH2 inhibition by GSK126 with confirmed induction of SRE binding factors and ABC transporters also led to an increase in oxaliplatin resistance in the exemplary cell lines HH and Jurkat (Appendix A). Additionally, EZH2-dependent regulation was ruled out as *SREBF1/2* and *ABCG1/2* expression was not consistently induced by EZH2 knockdown (Figure 4E).

Taken together, the analysis of mechanistic and transcriptional changes under pharmacological EZH2 inhibition indicate a universal and EZH2-independent mechanism by enhanced efflux that involves SRE binding proteins and ABC transporters.

### 3.3. SREBP and ABC Inhibition Reduce Effects Induced by Pharmacological EZH2 Inhibition

We hypothesized that SRE binding protein activation by pharmacological EZH2 inhibition led to activation of ABC transporters which results in increased oxaliplatin efflux (Figure 5D). As proof of principle, we used the SRE binding protein inhibitor Fatostatin as well as the ABC transporter inhibitor Glibenclamide in combination with oxaliplatin and pharmacological EZH2 inhibition by GSK126 in cell viability analysis of two cell lines with a mature T-cell lymphoma background, HH and DERL-2 (Figure 5A). In addition, two cell lines of immature T-ALL background, Jurkat and Loucy, were tested. While oxaliplatin resistance was still increased, the maximal reduction in toxicity was lowered by inhibition of either SRE binding proteins or ABC transporters using GSK126 (Figure 5B). This was further confirmed by the tendency of reduction in oxaliplatin resistance by the combinational incubation of SRE binding protein or ABC transporter inhibitors and EPZ6438 treatment using the exemplary cell lines HH, Jurkat, and Loucy (Appendix A). Neither Fatostatin nor Glibenclamide monotherapy were able to alter oxaliplatin sensitivity (Appendix A).

In further support, both the SRE binding protein and ABC inhibition were able to again increase the concentration of intracellular platinum that was lowered by pharmacological EZH2 inhibition (Figure 5C). Thereby, the level of intracellular platinum of oxaliplatin single therapy was regained. Again, neither Fatostatin nor Glibenclamide alone were able to influence the intracellular platinum content compared to oxaliplatin monotherapy (Appendix A).

## 4. Discussion

In the present study, we have revealed that pharmacological EZH2 inhibition is not suitable in combination with chemotherapeutic oxaliplatin which is commonly used in the second-line treatment of mature T-cell lymphomas. Our findings underline the need for complex investigations when combining new inhibitors with an established regimen as off-target effects of new inhibitors and potential antagonistic effects need to be taken into consideration.

Pharmacological EZH2 inhibition was proven to be effective in the treatment of entities with activating EZH2 mutations such as Diffuse Large Cell B-Cell as well as Follicular Lymphoma in phase II clinical trials [41,42]. However, malignancies with unmutated but overexpressed EZH2, such as T-cell leukemia, could also be suitable for EZH2 inhibition [43]. In an earlier study, we could show that mature T-cell lymphomas also showed EZH2 overexpression which impacts on patient outcomes [12]. We could validate these findings using an extensive gene expression profiling data set as well as by further immunohistochemistry evaluations in the current study. This underlines the rationale for EZH2 targeting in T-cell lymphomas. Currently, there are 30 clinical trials investigating EZH2 targeting in different entities registered on ClinicalTrials.gov (3 January 2023, search terms “EZH2” and “Recruiting”, “Active, not recruiting”, or “Enrolling by invitation”; Source: National Library of Medicine). Accordingly, the EZH2-specific inhibitor EPZ6438 was already approved for the treatment of epithelioid sarcoma by FDA [44]. As EZH2 is also overexpressed in T-cell leukemia and lymphomas, its inhibition is of interest in these entities and the effective but limited reduction in cell viability as single therapy was proven repeatedly but could not be validated in the current study [4,19]. This shows that influences of the particular cellular background need to be taken into consideration when using protein specific inhibitors as single agents.

Therefore, in clinical therapies, new drugs are often combined with established regimen in search of synergistic effects. However, antagonistic modulations of drug sensitivity also need to be taken into consideration. In the present study, we have shown that EZH2 inhibitors have moderate effects on the sensitivity modulation of the commonly used chemotherapeutics doxorubicin, cytarabin, and 5-azacytidin. Moreover, we have proven that pharmacological EZH2 inhibition by SAM-competitive small molecule drugs increase oxaliplatin resistance drastically in mature T-cell lymphoma, T-cell leukemia, and B-cell lymphoma cell lines. We have shown for HH, DERL-2, Jurkat, and Loucy that secondary readouts such as cell cycle deregulation and induction of apoptosis by oxaliplatin were reduced by additional pharmacological inhibition. While the initial response to oxaliplatin varied between the cell lines, the reduction in oxaliplatin activity seems to be universal indicating a mechanism of resistance independent from the primary oxaliplatin effect.

Additionally, the intracellular platinum content was lowered in HH and Jurkat by pharmacological EZH2 inhibition. This indicates that an increase in oxaliplatin efflux might be one of the mechanisms involved in the increase in resistance.

The effect was not correlated to the cellular background or EZH2 expression which demonstrates a universal mechanism independent from EZH2 inhibition. This was further underlined by the lack of induced oxaliplatin resistance by non-SAM-competitive EZH2 inhibitor DZNep and EZH2 knockdown or the lack of reduced intracellular platinum under EZH2 knockdown. SAM-competitive EZH2 inhibitors might also impede the activity of the related protein EZH1 that therefore needs to be taken into consideration as a possible off-target. In the setting of EZH2 knockdown, EZH1 would not be affected thereby potentially compensating for the induction of resistance. It was shown that there is an overlap of targets between the two proteins and EZH1 might act as a backup protein in case of EZH2 depletion [3,45].

However, off-target effects independent from PRC2 signaling also need to be taken into consideration. Lately, EZH2 independent effects of SAM-competitive EZH2 inhibitors have been shown in hepatocellular carcinoma cell lines and correlated with an induction of expression of the transcription factor SRE binding protein [39]. Furthermore, it was proven that ABCG1 transporter protein expression is under control of SRE binding protein [46]. ABC transporters are a family of proteins that are involved in cholesterol homeostasis and play a significant role in multidrug resistance [47]. ABCG1 upregulation was linked to oxaliplatin resistance in hepatocellular carcinoma while ABCG2 is known for promoting the efflux of a variety of substrates including cytotoxic agents [48,49]. Consistently, we found an upregulation of SRE binding factors 1 and 2 as well as ABCG transporters 1 and 2 by pharmacological EZH2 inhibition using SAM-competitive EZH2 inhibitors but not by non-SAM-competitive EZH2 inhibitor DZNep and EZH2 knockdown. Again, this indicates an EZH2 independent mechanism as both EZH2 inhibition by another mechanism of action and the loss of EZH2 did not lead to the same upregulation of genes, as seen for the tested SAM-competitive EZH2 inhibitors. While both GSK126 and EPZ6438 were able to induce the expression of SRE binding factors 1 and 2 as well as ABCG transporters 1 and 2, the effects were more profound using GSK126 treatment while EPZ6438 treatment at least showed tendencies. However, these differences demonstrate the deviation of the inhibitors’ efficacy, as also the induction of oxaliplatin resistance by EPZ6438 is overall smaller than by GSK126 treatment.

As the induction of oxaliplatin resistance was found independent of EZH1, EZH2, or H3K27 trimethylation status of the different cell lines, a PRC2-dependent, or non-canonical EZH2-signaling pathway is unlikely. We hypothesize that the block of oxaliplatin activity is indeed an EZH2-independent off-target effect of SAM-competitive EZH2-inhibitors. We propose that the SREBP1/2-mediated upregulation of ABCG1/2 leads to an increased oxaliplatin efflux which results in resistance. We have provided proof by the reduction in GSK126-induced oxaliplatin resistance and efflux by additional pharmacological SRE binding protein or ABC transporter inhibition. As neither Fatostatin nor Glibenclamide as monotherapy were able to alter oxaliplatin sensitivity or the intracellular platinum content, we hypothesize that the induction of SRE binding proteins or ABC transporters by pharmacological EZH2 inhibition is needed for the consecutive inhibitors to show effects.

However, the complexity of the investigated system needs to be taken into consideration. The shown data provide insight into the regulated mechanisms by pharmacological EZH2 inhibition and their effects on oxaliplatin sensitivity. We could prove that EZH2-independent SREBP1/2 and ABCG1/2 deregulation are involved in increasing oxaliplatin resistance. EZH2-dependent effects as well as other modulations were not investigated further and go beyond the scope of this paper.

## 5. Conclusions

While EZH2 inhibitors show promising however moderate results as single agents in T-cell lymphomas as well as synergistic effects with other chemotherapeutics, our data do not support the combination with platinum derivates. This study demonstrates that pharmacological EZH2 inhibition can lead to an increase in chemotherapy resistance, here shown for oxaliplatin, a common second-line treatment drug in T-cell lymphomas. This underlines the necessity of encompassing investigations of rational drug combinations.

## Figures and Tables

**Figure 1 cancers-15-03043-f001:**
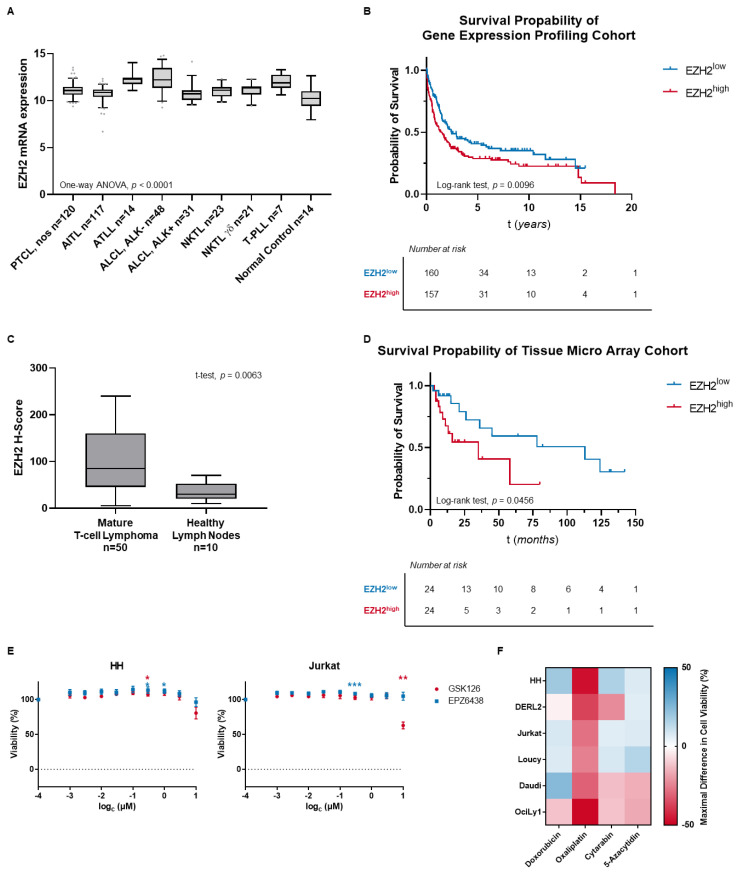
EZH2 is overexpressed in tumor tissue which is associated with prognosis. (**A**) In a cohort of MTCL with 381 cases, EZH2 mRNA expression was compared to normal lymphoid tissue (*n* = 14) by one-way ANOVA testing. (**B**) In the same cohort, survival analysis was performed by log-rank test. (**C**) In a tissue micro array of 50 MTCL cases, EZH2 protein expression was compared to healthy lymphoid tissue (*n* = 10) by unpaired Student’s *t*-test. (**D**) In the tissue micro array cohort, survival analysis was applied using log-rank test. (**E**) Cell viability response to SAM-competitive EZH2 inhibitors GSK126 and EPZ6438 was assessed in the cell lines HH and Jurkat (* *p* ≤ 0.05; ** *p* ≤ 0.005; *** *p* ≤ 0.001). (**F**) In a screening approach, established chemotherapeutics were used in combinational incubation with GSK126 and changes in sensitivity compared to chemotherapeutic single therapy were assessed.

**Figure 2 cancers-15-03043-f002:**
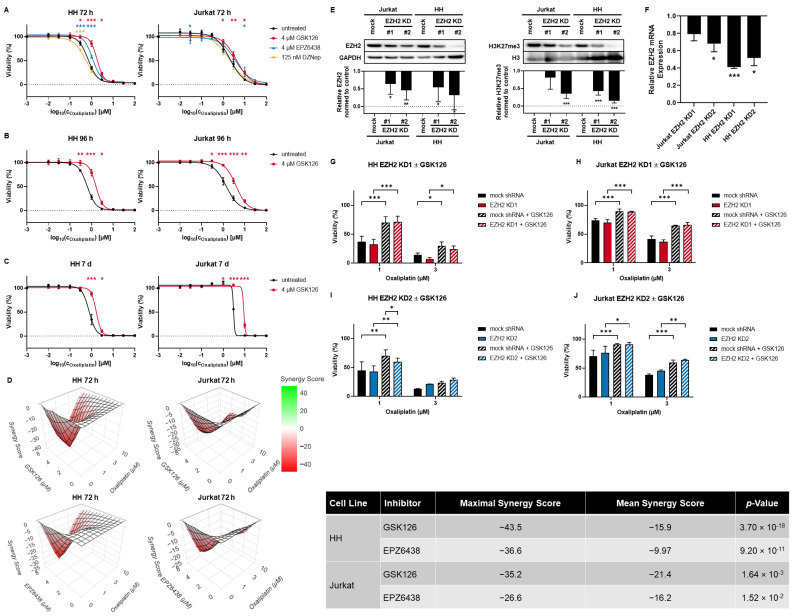
SAM-competitive EZH2 inhibitors increase oxaliplatin resistance significantly. Oxaliplatin resistance was induced by pharmacological EZH2 inhibition after (**A**) 72 h and persisted after (**B**) 96 h and (**C**) 7 d. Significant differences between oxaliplatin monotherapy and in combination with GSK126 (red) EPZ6438 (blue), and DZNep (yellow) are indicated by the respective colored asterisks. Each point represents mean ± SEM, *n* = 3 except for the respective 72 h of incubation, untreated data points which represent mean ± SEM, *n* = 6. (**D**) Synergy matrices of oxaliplatin in combination with GSK126 (**top**) or EPZ6438 (**bottom**) in HH and Jurkat were composed using Synergy Finder [26]. The HSA model was used for calculation. (**E**) EZH2 knockdown was established using shRNAs and validated in the cell lines HH and Jurkat by Western blot with the respective quantitative analysis shown below. Each bar represents mean ± SD, *n* = 5. (**F**) EZH2 mRNA reduction was validated by qPCR. Quantitative Western blot and qPCR analyses were normed to the respective mock control. Each bar represents mean ± SEM, *n* = 3. Modulation of chemotherapy sensitivity was not found in either HH (**G**,**I**) or Jurkat (**H**,**J**) by EZH2 knockdown. The induction of resistance can be regained by additional GSK126. Each bar represents mean ± SEM, *n* = 3 (* *p* ≤ 0.05; ** *p* ≤ 0.005; *** *p* ≤ 0.001; # number).

**Figure 3 cancers-15-03043-f003:**
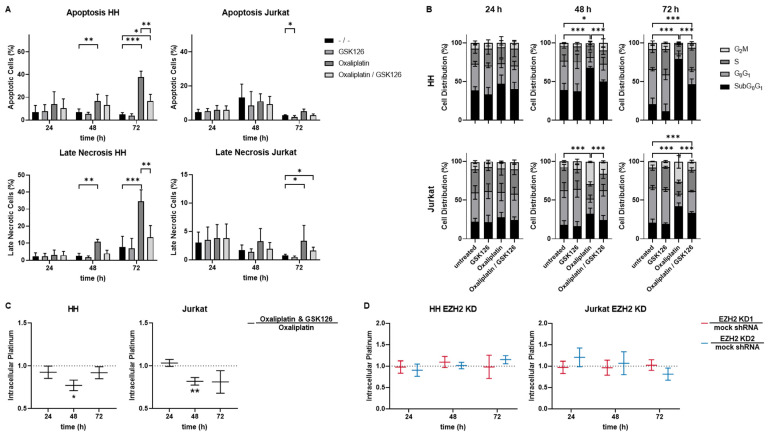
Mechanistic and molecular changes under pharmacological EZH2 inhibition. Pharmacological EZH2 Inhibition reduces the oxaliplatin induced (**A**) apoptosis and necrosis as well as (**B**) cell cycle deregulation. Each bar represents the mean ± SD, *n* = 3 (* *p* ≤ 0.05; ** *p* ≤ 0.005; *** *p* ≤ 0.001). (**C**) Intracellular Platinum concentration was measured by ICP-MS uncovering decreased intracellular platinum content by pharmacological EZH2 inhibition. (**D**) EZH2 knockdown did not reduce the intracellular platinum content normed to the respective mock control. Each line represents the mean ± SEM, *n* = 3 (* *p* ≤ 0.05; ** *p* ≤ 0.005; *** *p* ≤ 0.001).

**Figure 4 cancers-15-03043-f004:**
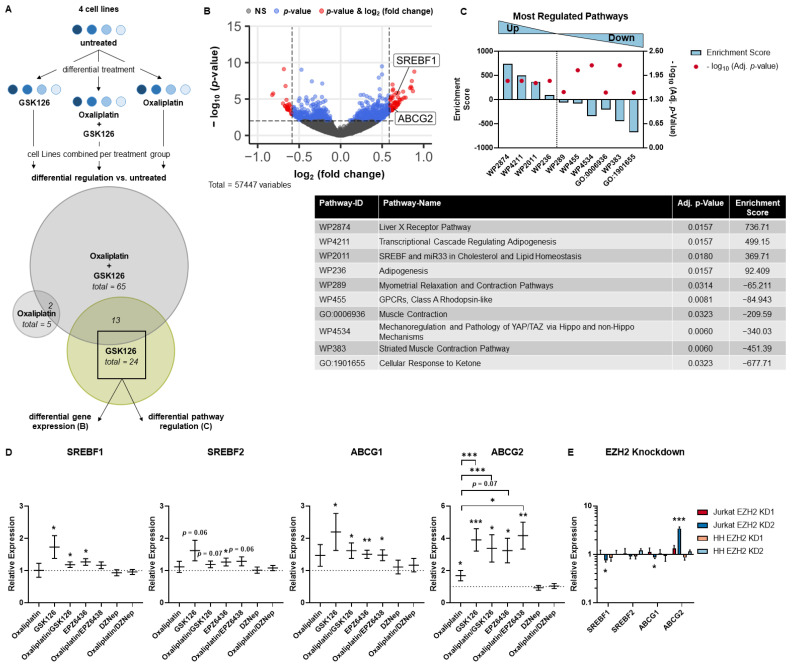
Transcriptional changes under pharmacological EZH2 inhibition. (**A**) Experimental setup for mRNA-sequencing analysis from the cell lines HH, DERL2, Jurkat, and Loucy combined including Venn analysis. Differentially expressed genes compared to untreated are indicated in italic. Further analyses were conducted from genes regulated by GSK126 treatment (green background). (**B**) Differentially expressed genes by pharmacological EZH2 inhibition using GSK126 compared to oxaliplatin. (**C**) Enriched and depleted pathways by pharmacological EZH2 inhibition using GSK126 compared to oxaliplatin. Pathway IDs with adjusted *p*-value and enrichment score are described below. (**D**) The sequencing findings for GSK126, oxaliplatin, and oxaliplatin/GSK126 treatments in relation to untreated control were validated by qPCR and expanded for EPZ6438, oxaliplatin/EPZ6438, DZNep, and oxaliplatin/DZNep treatment. Each line represents the mean from the four cell lines HH, DERL2, Jurkat, and Loucy ± SEM, *n* = 3 per cell line (* *p* ≤ 0.05; ** *p* ≤ 0.005; *** *p* ≤ 0.001). (**E**) EZH2 knockdown did not increase *SREBF1/2* and *ABCG1/2* expression consistently. Each bar represents mean ± SEM, *n* = 3.

**Figure 5 cancers-15-03043-f005:**
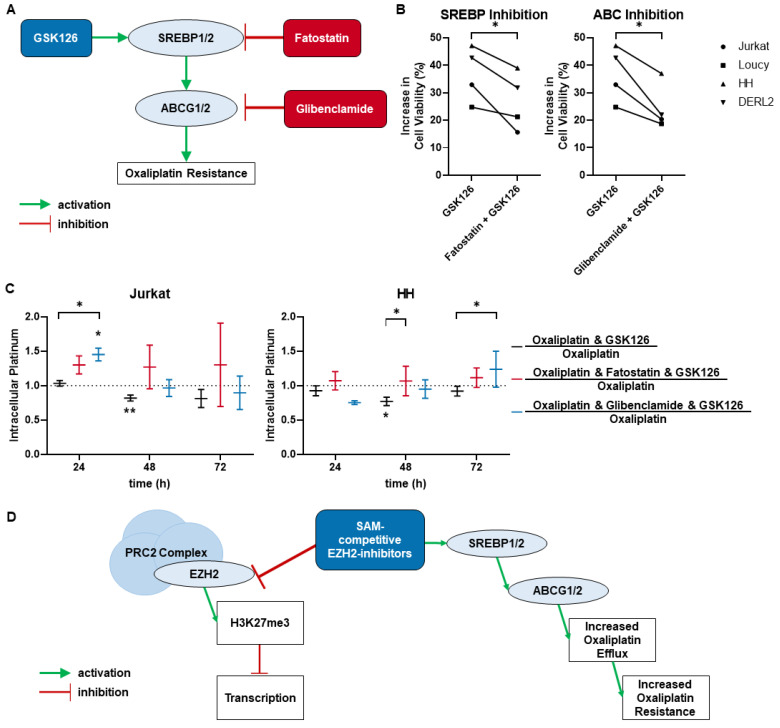
SREBP and ABC inhibition reduce oxaliplatin resistance and efflux induced by pharmacological EZH2 inhibition. (**A**) Setup for combinational approach of EZH2 inhibition and either inhibition of SRE binding proteins (Fatostatin) or ABC transporters (Glibenclamide). (**B**) In oxaliplatin viability assay, the inhibition of SRE binding proteins or ABC transporters was able to lower the induced oxaliplatin resistance by pharmacological EZH2 inhibition. Each point represents the mean of *n* = 3. (**C**) SREBP and ABC inhibitors reduce the decrease in intracellular platinum by pharmacological EZH2 inhibition. Each line represents mean ± SEM, *n* = 3 (* *p* ≤ 0.05; ** *p* ≤ 0.005). (**D**) Combined hypothesis of the uncovered mechanisms responsible for the induction of oxaliplatin resistance by pharmacological EZH2 inhibition. EZH2 mechanism adapted from Richly et al. [40].

## Data Availability

The raw mRNA sequencing data were generated by Genewiz (Leipzig, Germany) and included in the article (Appendix A). All further data were generated by the authors and are available on request. Original Western blots can be found in Appendix A.

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
