# Peer review of "SAM-Competitive EZH2-Inhibitors Induce Platinum Resistance by EZH2-Independent Induction of ABC-Transporters"

_cancers, 2023, doi:10.3390/cancers15113043_

Round 1
Reviewer 1 Report (Previous Reviewer 1)
The authors have made significant changes to the manuscript and improved the overall understanding and quality of the paper.
Now with this updated and revised manuscript, it is in line with the authors intention and their scientific hypothesis and has all relevant results presented in a clear and understandable manner.
I suggest the manuscript can fit in to the cancers journal and recommend for publication.
Good Luck!.
Please check minor syntax errors.
Author Response
Comment 1: “Please check minor syntax errors.”
We have corrected spelling and syntax errors as well as minor wording adjustments. The corrections are marked in MS Word track change mode.
Reviewer 2 Report (Previous Reviewer 3)
The authors demonstrate that EZH2 inhibition may be effective for the T-cell lymphoma, while combinational treatment with EZH2 inhibitors leads to platinum resistance induced by ABC transporters.
The EZH2-independent induction of SREBP1/2 by SAM-competitive EXH2-inhibitors may be described in the discussion more clearly. SAM in the title should be explained in the title or abstract.
Author Response
Comment 1: “The EZH2-independent induction of SREBP1/2 by SAM-competitive EXH2-inhibitors may be described in the discussion more clearly.”
Comment 2: “SAM in the title should be explained in the title or abstract.”
We have included the description of the abbreviation as well as the general work mechanism of the inhibitors shortly in the abstract. The corresponding edited text can be found on page 1, lines 43-45.
All links refer to the track change version of the manuscript (cancers-2406836-revision1-TC“).
This manuscript is a resubmission of an earlier submission. The following is a list of the peer review reports and author responses from that submission.
Round 1
Reviewer 1 Report
Comments and Suggestions for the Authors.
MS ID: cancers-2209059
Article.
Title: SAM-Competitive EZH2-Inhibitors Induce Platinum-Resistance by EZH2-Independent Induction of ABC-Transporters.
Authors: Elisabeth Groß et al.
In this manuscript “SAM-Competitive EZH2-Inhibitors Induce Platinum-Resistance by EZH2-Independent Induction of ABC-Transporters " by Elisabeth Groß et al. Authors study the effect of SAM-Competitive EZH2 inhibitor treatment in T-cell lymphomas increase the oxaliplatin resistance by increasing the expression of drug efflux pump ABCG2 a well characterized chemoresistance marker in several cancers. I appreciate the critical thinking of the authors in framing their idea of how combination therapy can affect the synergy and can cause negative effects than the expected positive outcome. Even though the authors put a great effort in compiling the data based on their hypothesis testing and the observations gathered, there are several areas where it is not very clear why and how the authors concluded the results in one segment of the study led to the following tests and observations. There are several key features missing and not very well explained in the results and discussion sections. Following are few key and important things.
1. It is not clear how the authors selected single working concentration of the drugs used in several cell lines without performing the dose response in each cell line. Like (4 µM GSK126, 4 µM EPZ6438, 125 nM DZNep, 1 µM Fatostatin and 30 µM Glibenclamide).
2. As the title claims the SAM-competitive EZH2 inhibitors, authors must have shown at least two SAM-competitive EZH2 inhibitors in the study to strengthen their claim. (Like JQEZ5 and PF06726304 acetate).
3. Figure 2 shows the Apoptosis and cell cycle data along with the intracellular platinum. However the inhibitor used by authors is well known to induce autophagy than the apoptosis which is reported previously and key references missing. Even though the authors show apoptotic and cell cycle data, the visual FACS plots are primarily needed to support the percentages shown in the bar plots.(Late necrosis is misleading with late apoptotic or necroptosis). Why is HH cell line shows very high baseline subG0G1, can be culture issue.
4. Figure 3 RNA Seq data is not very clear why there are hardly any gene signature that are more than 1fold change expression and how and why authors selected the four markers which are embedded along with several genes and not outstanding from other gene signatures and how and why they further continued on developing the story based on these observations is not very clear. It is misleading in this figure and in the explanation which cell line is used for this study. Because ABCG2 expression in HH is not prominent in western data shown.
Overall the study need more valid data points, controls, comparisons and in vivo data to support the author’s strong claim on SAM-competitive EZH2 inhibitors induce platinum resistance. And the scenarios can be different in different cancers.
At this level I think it is not recommended for the publication in cancers journal.
Reviewer 2 Report
The submitted paper by Elisabeth Gross and coauthors describe the off-target effect of SAM-competitive EZH2 inhibitors. They provided evidence on the protective impact of these compounds on the oxiplatin cytotoxicity.
Limitations:
1. if the studied agents are supposed to impact oxiplatin toxicity by inducing SREBP-dependent overexpression of ABCG2, then these should be added to cell culture before platinium drug to allow for transcriptional activation of ABCG2 and SREBP. Does platinium drug accumulate to similarly in cells treated with and without SAM inhibitors before ABCG2 overexpression (at the protein level) occurs?
2. for mechnistical interconnections between SAM-inhibitors, SREBP and ABCG2 more direct approaches must be taken. For example, glibenclamid is not a specific inhibitor of ABCG2, even elacridar inhibits both ABCG2 and ABCB1. Hence, silencing of ABCG2 and SREBP should replace the currently presented results. Ideally, authors should consider transient target silencing with siRNA to avoid compensation effect. The necessary experiments include:
a) comparing oxiplatin toxicity between siCTRL. siSREBP, siABCG2 with and without SAMs
b) comparing SAM-induced overexpression of ABCG2 between siCTRL and siCREBP
Reviewer 3 Report
The study demonstrates that EZH2 inhibitors induce platinum-drug resistance in cancer cells. The drug resistance is very important in cancer combination therapeutics. In Conclusions, the effect of EZH2 inhibitors on drug resistance may be emphasized more, rather than the synergistic effects of EZH2 inhibitors in T-cell lymphomas.